# PASTEL: PANORAMIC ALIGNMENT FOR MONOCULAR 4D RECONSTRUCTION

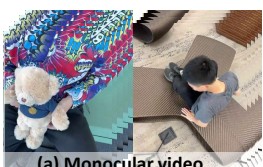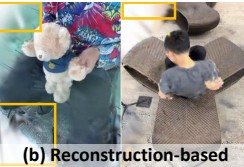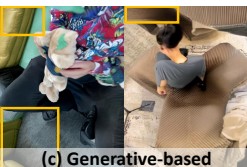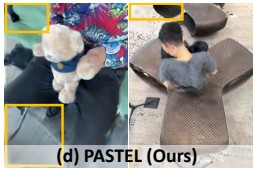

(a) Monocular video  (b) Reconstruction-based  (c) Generative-based  (d) PASTEL (Ours)

Figure 1: **Comparison of PASTEL with previous methods**. (a) PASTEL targets at 4D scene synthesis from monocular videos. (b) Reconstruction-based methods, such as Shape of Motion (Wang et al., 2024), can perform novel view synthesis. However, these methods struggle with inference of occluded and unobserved regions, leading to missing content in the yellow boxes. (c) Generative-based approaches, such as TrajectoryCrafter (YU et al., 2025), can infer unobserved regions in the yellow boxes. However, these approaches encounter challenges in generating consistent 4D scenes, as shown in the distorted generated results. (d) In comparison, PASTEL shows seamless extension of rendering beyond observable boundaries in the yellow boxed areas. At the same time, PASTEL preserves high fidelity and consistency with the original monocular videos.

## ABSTRACT

Reconstructing 4D scenes from casually captured monocular video is vital for applications in virtual reality (VR) and embodied AI. Recent advances in 4D reconstruction and novel view synthesis have significantly propelled this capability. However, all existing methods depend on pixel-level supervision and cannot recover regions beyond visible camera limits. Consequently, we introduce a new paradigm that reconstructs both the "visible regions" from monocular input and the "invisible regions" beyond observable camera boundaries. The most intuitive solution is to leverage video generation models as "generative priors". However, their inherent stochasticity and large solution space prevent stable and consistent view synthesis. As a result, naively incorporating generative content into the reconstruction process often causes artifacts, especially when camera trajectories deviate significantly from the original video. To overcome these challenges, we present Panoramic Alignment for StraTegic ExpLoitation of Generative Priors (**PASTEL**) to mitigate inconsistencies brought by generative priors. Specifically, PASTEL first aligns the scene into a spherical panoramic representation. Within this space, it identifies trajectories that minimize deviation while maximizing exploration beyond observable boundaries. These trajectories enable generative priors for stable and consistent exploration beyond visible camera limits. Experimental results show that PASTEL can not only extrapolate plausible scene content beyond the observable boundaries of input monocular videos, but also substantially boost monocular 4D reconstruction performance, outperforming the previous state-of-the-art method by 1.2dB in PSNR on the DyCheck Iphone dataset.

## 1 INTRODUCTION

Reconstructing geometry-consistent and motion-coherent 4D scenes from monocular video (Li et al., 2023; 2021) remains a fundamental and unsolved challenge in computer vision. Recent advances in novel view synthesis and dynamic scene reconstruction (Park et al., 2021a; Luiten et al., 2024; Wu et al., 2024a) have enabled progress in modeling motion dynamics within multi-view settings (Wang

et al., 2024; Lei et al., 2025). However, these approaches face substantial limitations when extended to monocular inputs. This is because monocular videos usually suffer from limited fields of view and severe occlusions. Meanwhile, reconstruction-based methods rely on pixel-level supervision, which cannot infer regions beyond visible camera limits, as shown in Figure 1 (b). The absence of such "invisible regions" introduces a significant gap for novel view synthesis, which significantly limits the immersive quality and user experience of the reconstructed scenes.

To tackle these "invisible regions", our key insight is to effectively leverage camera-controlled generative models as "generative priors" to infer regions beyond observable camera boundaries. Recent research on camera-controlled video generation (YU et al., 2025; You et al., 2025; Bai et al., 2025b;a) demonstrates strong capabilities for novel view synthesis. However, directly applying these methods to reconstruction introduces notable limitations. Specifically, the inherent randomness within the diffusion denoising process often introduces deviations in the detailed texture and even the overall color. This deviation often results in inconsistencies and artifacts when directly applied to the 4D scene reconstruction, as exhibited in Figure 1 (c). The greater the trajectory deviates from the original video, the more content needs to be inpainted by the generative prior, resulting in more severe artifacts. Therefore, designing effective camera trajectories is crucial. Optimal camera trajectories must maximize exploration beyond observable regions while minimizing both trajectory deviation and overlap. However, a trajectory with $L$ camera positions has $L \times 4 \times 4$ degrees of freedom, since each camera is defined by a $4 \times 4$ extrinsic matrix. This makes direct search of optimal cameras highly challenging in cartesian coordinate.

To address this challenge, we propose Panoramic Alignment for Strategic Exploitation of Generative Priors (**PASTEL**). Panorama representations naturally support perspective exploration (Szeliski & Shum, 1997; Zomet et al., 2006), so we leverage this property to design optimal trajectories for 4D scene exploration. Our key insight is to transform the 4D scene reconstruction into a panoramic space. With this representation, we no longer need to design high-dimensional camera extrinsics in full 3D Cartesian space, which often leads to suboptimal coverage and cross-view inconsistencies. Instead, viewpoint planning becomes a tractable 2D directional search. Within this unified panoramic domain, we design expansion trajectories that minimize viewpoint distortion and maximize spatial coverage beyond the observable boundary. This design guarantees consistent novel-view synthesis with high exploration efficiency. The resulting panoramic trajectories provide warped conditioning inputs to a generative prior, which inpaints invisible regions for scene expansions. These generated results then provide targeted supervision for strategic 4D reconstruction, improving both completeness and cross-view consistency.

To summarize, we make the following contributions: (1) We identify and highlight a long-neglected yet critical issue in monocular 4D synthesis: the inability to synthesize invisible regions in the input video. To address this, we introduce PASTEL, a groundbreaking framework to address these regions beyond observable boundaries for novel view synthesis. (2) We propose panoramic scene alignment, a new representation that reformulates scene exploration as a 2D directional trajectory search. This enables efficient, interpretable trajectory planning beyond observable boundaries. (3) PASTEL consistently outperforms both reconstruction- and generative-based baselines across diverse datasets and scenarios. PASTEL shows seamless extension of rendering capabilities beyond monocular video coverage while maintaining high fidelity and consistency, thereby establishing new standards in 4D synthesis in challenging visual scenarios.

## 2 RELATED WORKS

**Reconstruction from monocular video** Video is one of the richest resources available on the Internet. Creating immersive experiences and constructing virtual worlds from monocular video holds significant value for both users and many downstream industries. To address this challenge, Lei et al. (2025) integrates prior knowledge from various 2D foundation models to optimize the proposed motion scaffold representation. Wang et al. (2024) exploits the low-dimensional structure inherent in scene motion to constrain the full-sequence-long 3D motion of each point. Li et al. (2023) combines Transformer architectures and image-based rendering to achieve photorealistic novel view synthesis from monocular videos. Zhang et al. (2025); Han et al. (2025); Sucar et al. (2025) employ a data-driven approach, training models to predict the geometry and motions from stereo image pairs. Some other works extend Gaussian Splatting into dynamic view synthesis. They often incorporate

regularization through MLPs (Yang et al., 2024) or low-rank motion representations (Huang et al., 2024). However, these methods rely on pixel-level supervision to reconstruct 3D or 4D scenes. Consequently, they cannot infer "invisible regions" where no ground truth is available for supervision.

**Camera-controlled video generation** Spurred by the recent advancement of AIGC, researchers have begun to explore using video generative models for directly synthesizing observations from the altered camera trajectories, which can leverage the strong prior knowledge of generative models to adequately constrain the complexity of this task. Among them, You et al. (2025); YU et al. (2025); Zhang et al. (2024) condition the video diffusion models with novel viewpoint cloud rendering. In contrast, Zhou et al. (2025); He et al. (2024; 2025) discard the explicit geometry estimation by only conditioning the model with the plücker embedding of target views. Bai et al. (2025b;a) adopt the Diffusion Transformer (DiT) (Peebles & Xie, 2023) architecture with view or frame attention, generating videos from multiple fixed views or a specified trajectory. Despite the remarkable success in view synthesis, these methods typically exploit denoising process from diffusion models for video generation, which contain severe randomness. This randomness usually leads to severe conflict and inconsistency when directly applied to 3D/4D scene reconstruction.

**Panoramic representations** Classical approaches assume the panoramic scene as a static image plane. Szeliski & Shum (1997); Zomet et al. (2006) propose cylindrical warping and seam matching to assemble multiple frames onto a unified cylindrical panoramic surface. Zaragoza et al. (2013); Zhang & Liu (2014) introduced parallax-tolerant image stitching processes, relaxing the strict planar assumption but still producing static panoramas. To further improve reconstruction of object surfaces, Zheng et al. (2007) developed layered depth panoramas, and Hedman et al. (2017); Hedman & Kopf (2018) introduced mesh-based representations for interactive rendering. Recently, Chugunov et al. (2024) proposed spherical neural light fields for implicit panoramic image stitching and re-rendering. These advances support vivid wide-angle rendering, but they remain limited by the lack of explicit geometry and physical constraints, making it difficult to represent dynamic object motion and interactions. In comparison, our work leverages panoramic representations to explore out-of-view regions through generative priors. We then distill all explorations into a 4D Gaussian representation, enabling geometrically consistency modeling of dynamic scenes.

# 3 METHODS

We propose PASTEL for high-quality 4D scene synthesis from a casually captured monocular video. This 4D scene synthesis includes unbounded regions beyond visible camera limits, which maintain coherence with input monocular videos. As shown in Figure 2, PASTEL first reprojects all video pixels into panorama space (Section 3.2). Then, the optimal trajectory is designed based on panorama representation to enable generative priors to produce consistent exploration of regions beyond observable boundaries (Section 3.3). The generated results of unobservable areas finally supervise the synthesis of the whole 4D scene to maintain high fidelity and consistency (Section 3.4).

## 3.1 PRELIMINARIES

**Camera-controlled video re-generation** Camera-controlled video re-generation aims to redirect the camera trajectory of input monocular video for creating immersive exploration. Because of the diffusion model's desirable diversity and scalability, it has become the *de facto* solution for the various visual generation tasks. Given a video $\mathcal{V}_0 \in \mathbb{R}^{T \times C \times H \times W}$, the forward diffusion process gradually adds noise on it as:

$$q(\mathcal{V}_t | \mathcal{V}_{t-1}) = \mathcal{N}(\mathcal{V}_t; \sqrt{1 - \beta_t} \mathcal{V}_{t-1}, \beta_t \mathbf{I}), \tag{1}$$

where $\beta_t \in (0, 1)$ represents the pre-defined variance level at time step $i$, $\mathcal{V}_1$ will degrade into pure Gaussian noise. The diffusion model is proposed to learn the reverse process, thus being able to sample from the data distribution by denoising pure Gaussian noise. The reverse process can be formulated as:

$$\mathcal{V}_{t-1} = \frac{1}{\sqrt{\alpha_t}} \left( \mathcal{V}_t - \frac{\beta_t}{\sqrt{1 - \bar{\alpha}_t}} \epsilon_\theta (\mathcal{V}_t, i) \right), \tag{2}$$

where $\bar{\alpha}_t = \prod_1^t (1 - \beta_t)$, $\epsilon_\theta$ is a noise estimator, modeled as a Diffusion Transformer (DiT). To achieve affordable computation for long and high-resolution video, the diffusion process is conducted

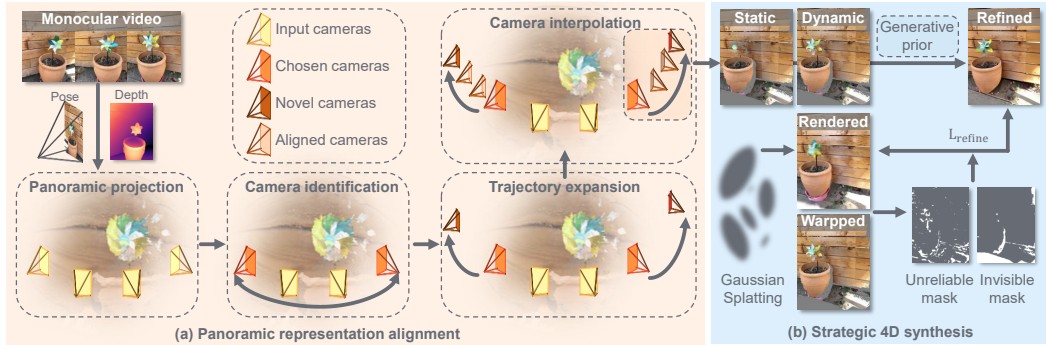

Figure 2: **Architecture overview of PASTEL**. (a) Given a monocular video with predicted camera poses and depth, PASTEL designs panoramic representation alignment by projecting all video frames into a unified panoramic space. PASTEL then devises chosen cameras serving as the beginning of trajectories. The trajectory is then expanded from the chosen camera along with our designated direction, and interpolated with multiple aligned camera viewpoints to ensure compatibility with generative priors. (b) Along the designated trajectory, both static and dynamic components are utilized to achieve a comprehensive view expansion, which feeds into the diffusion prior for refinement. Finally, we compare warped and rendered images from Gaussian Splatting to identify unreliable and invisible regions, which are used to strategically supervise the 4D scene.

in the latent space, which can be transformed between pixel space via a 3D VAE. Since our task is conditioned on the original trajectory video $\mathcal{V}'$, $\epsilon_\theta$ can also take $\mathcal{V}'$ as input by concatenating its warped version with the noisy latent and directly referring to it through cross attentions.

**Challenges**  Despite impressive perceptual quality, video diffusion alone is not sufficient for real-world applications that require real-time rendering and consistent outputs. This limitation motivates us to explore distilling video generation priors into a structured 4D scene representation. However, the inherent randomness of the diffusion denoising process often introduces deviations in both detailed textures and global colors. These deviations lead to inconsistencies and artifacts when directly used for 4D scene reconstruction, which become extremely severe when the synthesized camera trajectories deviate strongly from the original video. Consequently, designing effective camera trajectories is crucial. The trajectories should expand exploration beyond the observable regions while minimizing both deviation and overlap. However, trajectory optimization in the cartesian coordinate is difficult: a trajectory with $L$ camera positions involves $L \times 4 \times 4$ degrees of freedom, since each camera is parameterized by a $4 \times 4$ extrinsic matrix. This high dimensionality makes direct search for optimal trajectories intractable in the cartesian coordinate.

## 3.2  PANORAMIC SCENE REPRESENTATION

Naively, trajectory discovery is performed in full 3D Cartesian space. For $M$ trajectories, each with $L$ viewpoints and full 4×4 extrinsic matrices, the system must optimize $M \times L \times 4 \times 4$ parameters. This high-dimensional search makes it extremely difficult to design expansion paths that are simultaneously efficient, non-overlapping, and geometrically consistent. In practice, such optimization often leads to suboptimal scene coverage and visible inconsistencies due to redundant or poorly conditioned viewpoints. To address this challenge, PASTEL first converts the monocular video into a panoramic scene representation (Szeliski & Shum, 1997; Zomet et al., 2006). The resulting panoramic representation provides a global and unified view of the scene.

Given a monocular video of $T$ frames, we first back-project each frame into 3D space using our precomputed depth map, camera extrinsics, and intrinsics. This step produces a sequence of 3D point clouds, each aligned to the world coordinate system. We then compute the average of all 3D points as the spherical center of the panorama $P^c = (x^c, y^c, z^c)$. We then reproject all 3D points into the panoramic space. Concretely, for each point in point cloud $p_t^j = (x_t^j, y_t^j, z_t^j) \in P_t$, we first compute

the polar coordinates as:

$$\theta_t^j = \arctan\left(\frac{y_t^j - y^c}{x_t^j - x^c}\right), \phi_t^j = \arctan\left(\frac{z_t^j - z^c}{\sqrt{(x_t^j - x^c)^2 + (y_t^j - y^c)^2}}\right). \tag{3}$$

Finally, we compute the pixel coordinates within the panorama:

$$u_t^j = \left(\frac{\theta_t^j}{2\pi} + 0.5\right) \times w_p, \quad v_t^j = \left(0.5 - \frac{\phi_t^j}{\pi}\right) \times h_p, \tag{4}$$

where $w_p$ and $h_p$ are width and height of 2D panorama. This produces a dense 2D panorama that encodes the global scene. Meanwhile, for each image frame captured by the camera in timestamp $t$, we can also track all its image pixels through the projection. This could yield a projected point sets in the panorama for each image, which is denoted as $M_t = \{(u_t^j, v_t^j) \mid j = 1, ..., n_t\}$.

Compared with Cartesian coordinates, the panoramic representation collapses the high-dimensional trajectory parameterization into a compact spherical coordinate system. This not only makes trajectory optimization significantly more tractable, but also explicitly exposes global scene structure. Within this unified domain, PASTEL identifies expansion trajectories that minimize geometric distortion while encouraging generative priors to explore regions beyond the observable boundary, enabling consistent and efficient novel-view expansion.

### 3.3 Adaptive Trajectory Identification Under Panorama

After projecting the video into panoramic space, PASTEL seeks to determine camera trajectory under the panorama representation. The panoramic formulation collapses the original 6-DoF camera search into a compact 2D directional domain. This dimensionality reduction makes trajectory optimization well-posed and interpretable. The optimal trajectory should balance two goals: minimal deviation from the original cameras and maximal exploration of the unseen scene. Achieving this balance ensures that generative priors can produce high-quality and consistent inference of regions beyond visible camera limits. It will eventually guarantee more consistent synthesis for the final 4D synthesis.

To ensure uniform exploration with minimum overlap, we determine our trajectory direction $d_m$ through the evenly expanded angle. This is defined as:

$$d_m = (d_{m,u}, d_{m,v}) = \left(\cos\left(\frac{2\pi \cdot m}{M}\right), \sin\left(\frac{2\pi \cdot m}{M}\right)\right), \tag{5}$$

where $m \in \{1, ..., M\}$, $M$ is the total number of evenly expanded directions. Each $d_m$ represents a unit vector in the XY-plane corresponding to the angle $\frac{2\pi \cdot m}{M}$. For each direction, we choose a starting camera by finding the point in the panorama that extends furthest along $d_m$. In section 3.2, we have achieved the point sets $M_t = \{(u_t^j, v_t^j) \mid j = 1, ..., n_t\}$ for each image frame captured by the camera in timestamp $t$ under the panorama coordinate. Consequently, we can identify the chosen camera $c_m$ by finding the largest point in each point sets $M_t$ given the direction $d_m$. This is done by maximizing the inner product between the relative 2D panorama point sets and the direction $d_m$:

$$c_m = \arg\max_t \left\{\max_j \left((u_t^j - x^c)d_{m,x} + (v_t^j - y^c)d_{m,y}\right)\right\}. \tag{6}$$

It ensures that the trajectory begins at the edge of the visible region and naturally expands outward.

Then, we expand the camera trajectory from the chosen camera $c_m$ through the direction $d_m$ in the panoramic space. The shifted location is then mapped back into Cartesian space, yielding the position of the novel camera. The orientation of the novel camera is defined as the direction pointing from the novel camera position toward the object center. This design ensures that new viewpoints remain consistent with the scene structure while allowing efficient exploration beyond the observed region. After that, we create a trajectory by interpolating between the novel camera extrinsics $\tilde{R}_{t,m}$ and the original camera extrinsics $R_t$ through

$$R_{t,m}^{\theta_l} = (1 - \theta_l) \cdot R_t + \theta_l \cdot \tilde{R}_{t,m}, \tag{7}$$

where $\{\theta_l\}_{l=1}^L \in [0, 1]$ denotes the interpolation factor.

By calculating optimal trajectory directions through evenly expanded angles and interpolating between original and new camera positions, we create trajectories with minimum deviation and maximum exploration of areas beyond observable boundaries. This is made feasible by the panorama representation, which reduces the target trajectory's degrees of freedom from $L \times 4 \times 4$ to a compact directional search. This trajectory design serves as a principled prior for stabilizing generative models. It avoids abrupt viewpoint jumps, maintains global scene coherence, and ensures that novel views align smoothly with diffusion model expectations. As shown in our ablations, this stable viewpoint progression directly improves temporal consistency and reduces generative drift.

### 3.4 COMPREHENSIVE VIEW EXPANSION

After designing the optimal camera trajectory, we further leverage it to guide generative priors for 4D synthesis beyond visible camera limits. To ensure consistency of generated results and the 4D scene, we design a static-dynamic projection that aligns static regions from all video frames into the conditioning input of the generative prior. This alignment preserves coherence with the original video, thereby supporting the generation of more consistent videos.

Specifically, we first reproject the static regions of each frame into 3D point clouds $P_t^{\text{static}}$. The static areas are obtained by masking out moving objects using pre-computed dynamic masks from optical flow estimation. Next, we merge all static point clouds across time to form a denser and wider 3D coverage $\bigcup_{t=1}^{T} P_t^{\text{static}}$. This aggregated point cloud is then projected into each target trajectory view, producing static background images $I_t^{\text{static}}$. We then process the moving regions separately. Each frame, together with its depth map, is reprojected onto the target views as $I_t^{\text{moving}}$. This provides the dynamic components that align with the designed camera trajectory. Finally, we combine the static background and the reprojected moving objects as:

$$I_t^{\text{warped}} = I_t^{\text{static}} \cup I_t^{\text{moving}}, \tag{8}$$

where $t$ denotes different timestamps of the frame. In this view expansion design, static content from all timestamps can be merged into a unified, viewpoint-consistent background. It prevents the diffusion model from hallucinating inconsistent geometry and provides a stable global reference across the entire 4D sequence. The resulting warped frames form a trajectory-aligned video $\mathcal{V}^{\text{warped}} = \{I_t^{\text{warped}}\}_{t=1}^{T}$, which is applied as guidance for camera-controlled video re-generation. The generative prior then refines this warped video to produce the refined outputs $\tilde{\mathcal{V}}'$.

PASTEL further designs a strategic supervision approach to minimize the adverse effects of RGB estimation deviations from generative priors. The key for strategic supervision is to exploit synthesized videos $\tilde{\mathcal{V}}'$ and the the previously warped video $\mathcal{V}^{\text{warped}} = \{I_t^{\text{warped}}\}_{t=1}^{T}$ to optimize unreliable and invisible regions of the target 4D scene. Specifically, we identify invisible areas as areas not covered by $\{I_t^{\text{warped}}\}_{t=1}^{T}$. This design could achieve the invisible mask as:

$$M_t^{\text{invisible}} = \mathbf{1}\left(\{I_t^{\text{warped}}\}_{t=1}^{T} < 0\right). \tag{9}$$

We further detect unreliable areas by comparing the Structural Similarity Index (SSIM) between the rendered RGB images $\{I_t^{\text{render}}\}_{t=1}^{T}$ and the warped RGB images $\{I_t^{\text{warped}}\}_{t=1}^{T}$. We set areas where the SSIM falls below a threshold $\epsilon$ as unreliable to achieve the unreliable mask:

$$M_t^{\text{unreliable}} = \mathbf{1}\left(\text{SSIM}(I_t^{\text{render}}, I_t^{\text{warped}}) < \epsilon\right). \tag{10}$$

Finally, we optimize our target scene representation through both the invisible mask and the unreliable mask. The refinement loss is thus defined as:

$$L_{\text{refine}} = L_{\text{rgb}}(M_t^{\text{refine}} \cdot I_t^{\text{render}}, M_t^{\text{refine}} \cdot I_t^{\text{refined}}), \tag{11}$$

where $M_t^{\text{refine}} = M_t^{\text{invisible}} \cup M_t^{\text{unreliable}}$ denotes regions for refinement.

Through unreliable and invisible masks, we ensure that generative priors are specifically applied to regions that require refinement. This strategic supervision prevents diffusion priors from overwriting well-reconstructed regions. As a result, the generative priors are exclusively exploited to inpaint regions beyond the observable boundary in the Gaussian Splatting. It further avoids inconsistencies arising from generative priors which may artifact the final 4D scene.

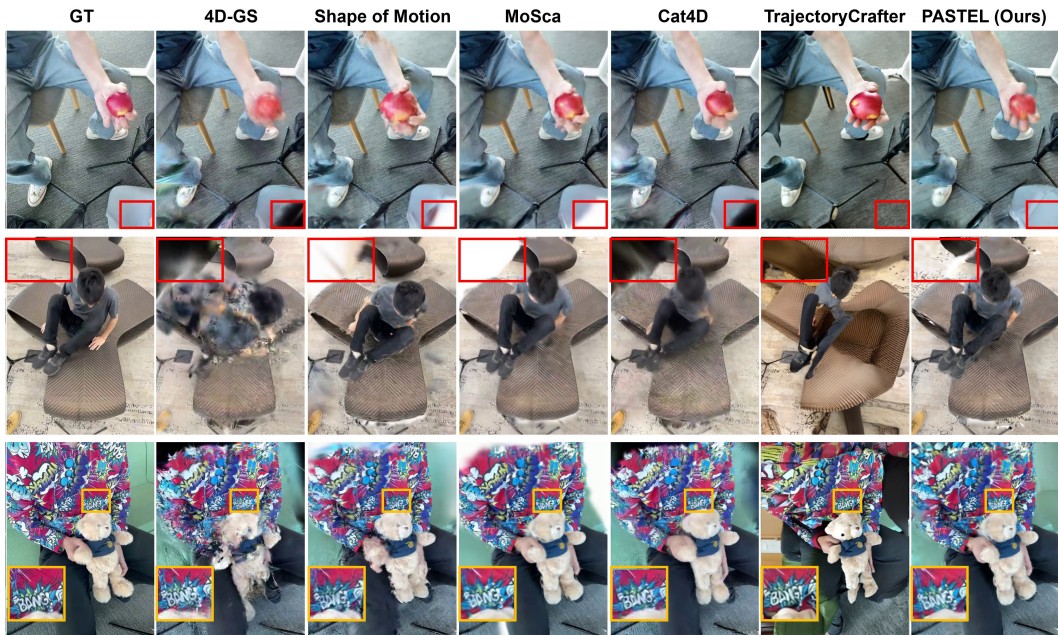

Figure 3: **Qualitative comparison on DyCheck (Li et al., 2023).** The red boxed areas highlight synthesis beyond covisible regions. Our method achieves superior overall quality and consistency with the ground truth, excelling in both reconstruction fidelity and the ability to induce details in areas beyond covisible regions.

Table 1: **Comparison with both reconstruction-based and generation-based methods on Dy-Check (Li et al., 2023).** 'Rec.' denotes reconstruction time in hours. "FPS" denotes rendering speed (frames per second). Our model significantly outperforms all previous reconstruction-based and generation-based methods in PSNR, SSIM, and LPIPS, with comparable time cost.

| Method | PSNR↑ | SSIM↑ | LPIPS↓ | mPSNR↑ | mSSIM↑ | mLPIPS↓ | FID↓ | Rec. | FPS |
|---|---|---|---|---|---|---|---|---|---|
| 4D GS (Wu et al., 2024a) | 15.71 | 0.450 | 0.398 | 16.54 | 0.594 | 0.347 | 127.52 | 1.2 | 44 |
| Shape-of-Motion (Wang et al., 2024) | 16.79 | 0.510 | 0.391 | 17.32 | 0.598 | 0.296 | 121.36 | 2.0 | 40 |
| MoSca (Lei et al., 2025) | 17.33 | 0.572 | 0.355 | 19.32 | 0.706 | 0.264 | 117.53 | 1.3 | 38 |
| Cat4D (Wu et al., 2024b) | 15.91 | 0.427 | 0.398 | 17.39 | 0.607 | 0.341 | 125.97 | - | - |
| TrajectoryCrafter (YU et al., 2025) | 12.71 | 0.324 | 0.519 | 14.24 | 0.417 | 0.519 | 137.30 | - | - |
| PASTEL (Ours) | **18.58** | **0.585** | **0.325** | **19.67** | **0.720** | **0.240** | **114.16** | 1.5 | 38 |

## 4 EXPERIMENTS

### 4.1 EXPERIMENTAL SETUP

**Datasets** To quantitatively assess our rendering outcomes both within and beyond the camera's line of sight, we compare our approach to existing methods using the currently most challenging dataset—the DyCheck iPhone (Li et al., 2023). This dataset provides covisible masks that encompass regions outside the range of all training cameras, making it ideally suited for a comprehensive evaluation of our performance. Furthermore, we conduct experiments on the NSFF dataset (Yoon et al., 2020) to measure the ability of our method with fewer frames. We further evaluate the generalization of PASTEL through reconstruction from real-world monocular videos in the Supplement.

**Metrics** Previous methods evaluated on the DyCheck iPhone dataset (Li et al., 2023) often focus only on areas within covisible masks, utilizing covisibility masked metrics such as mPSNR, mSSIM, and mLPIPS. However, examining occluded and out-of-view regions is crucial for enhancing user or robotic inference experiences in virtual reality and embodied AI contexts. Therefore, we also evaluate all methods, including areas beyond covisible masks to assess their ability to extend views outside the camera boundaries and ensure the completeness of 4D scenes, which are denoted as PSNR, SSIM, and LPIPS. Since the ground truth beyond covisible masks is unavailable in the training monocular videos, it is impractical to require high pixel-level fidelity. Therefore, we devised a novel benchmark to comprehensively evaluate the synthesis results at these views. Specifically, we employ the Fréchet

Figure 4: **Qualitative comparison on NSFF (Yoon et al., 2020).** Our method achieves better overall quality and consistency with the ground truth. Please zoom in for clearer visualization.

Inception Distance (FID) (Heusel et al., 2017) between synthesized views in novel trajectories and captured images from the original trajectories to assess the realism of synthesized images.

**Implementation details** Our panoramic representation is parameterized in a spherical coordinate system. The width and height of 2D panorama are set as $w_p = 2048$ and $h_p = 2048$. Each pixel corresponds to a 3D direction vector defined by the azimuthal angle $\phi \in [0, 2\pi)$ and polar angle $\theta \in [-\frac{\pi}{2}, \frac{\pi}{2}]$. This mapping adheres to the standard spherical-to-Cartesian conversion. For monocular video without depth, camera pose or dynamic mask, we employ UniDepth (Piccinelli et al., 2024) for depth estimation, BootsTAPIR (Doersch et al., 2024) for pose estimation, and RAFT (Teed & Deng, 2020) to estimate dynamic part through optical flow. Our generative prior is TrajectoryCrafter (YU et al., 2025), which includes 50 steps of inference. The 4D scene is represented through Motion Scaffolds (Lei et al., 2025), optimized through 6000 steps. We set $\epsilon = 0.3$ to achieve unreliable mask. The frame length (L) are set as 49 in accordance with (YU et al., 2025). The interpolation factors $\theta_l$ are evenly distributed within the range [0, 1] for smooth interpolation. We set the total number of expanded directions $M = 8$, which thoroughly covers all expanded regions with minimal overlaps.

## 4.2 COMPARISON WITH STATE OF THE ART

We compare the performance with both reconstruction-based methods and generative-based methods on the Dycheck iPhone dataset (Li et al., 2023), on the NSFF dataset (Yoon et al., 2020), and under other random real-world monocular videos.

On the Dycheck iPhone dataset (Li et al., 2023), the quantitative results are reported in Table 1, and qualitative results are shown in Figure 3. Reconstruction-based techniques typically excel in regions within the camera's field of view, showing competitive performance in covisibility masked metrics, such as mPSNR, mSSIM, and mLPIPS. However, these methods often lack the capability to interpret occluded regions or areas that fall outside the coverage of covisibile masks, with no prediction of areas beyond the covisibility masks, as illustrated in the red box parts in Figure 3. On the other hand, generation-based methods possess the ability to induce regions beyond the covisibility masks but may suffer from inconsistencies in illumination and color fidelity. This is attributed to the denoising processes inherent in generative models, which can introduce dataset-specific color biases that distort the original hues. In comparison, PASTEL demonstrates consistency with monocular video inputs while effectively extending its rendering capabilities to regions that fall outside the coverage of monocular videos in Figure 3. PASTEL thus achieves state-of-the-art performance across areas both within and beyond the covisibility mask in Table 1, outperforming the previous state of the art (Lei et al., 2025) by 1.2dB in PSNR. Moreover, PASTEL achieves the best overall quality on the NSFF

Table 2: **Comparison with reconstruction-based methods on Nvidia (Li et al., 2023).** The best results are highlighted in bold, while the second best results are underlined.

| Method | PSNR↑ | LPIPS↓ | Method | PSNR↑ | LPIPS↓ |
|---|---|---|---|---|---|
| D-NeRF (Pumarola et al., 2021) | 21.49 | 0.232 | CTNeRF (Miao et al., 2024) | 26.13 | 0.082 |
| NR-NeRF (Tretschk et al., 2021) | 19.69 | 0.323 | DynPoint (Zhou et al., 2024) | 26.53 | 0.068 |
| TiNeuVox (Fang et al., 2022) | 19.74 | 0.285 | D-NPC (Kappel et al., 2024) | 25.64 | 0.109 |
| HyperNeRF (Park et al., 2021b) | 17.60 | 0.367 | RoDynRF (Liu et al., 2023) | 25.89 | **0.067** |
| NSFF (Li et al., 2021) | 24.33 | 0.199 | Casual-FVS (Lee et al., 2023) | 24.57 | 0.081 |
| DynNeRF (Gao et al., 2021) | 26.10 | 0.082 | GaussianMarbles (Stearns et al., 2024) | 22.32 | 0.129 |
| MonoNeRF (Tian et al., 2023) | 25.62 | 0.106 | MoSca (Lei et al., 2025) | 26.72 | 0.070 |
| 4DGS (Wu et al., 2024a) | 21.45 | 0.199 | PASTEL (Ours) | **26.75** | 0.068 |

Table 3: **Ablation study on different components of the system.** Panoramic representation, trajectory alignment, and view expansion are crucial in views beyond camera range. Strategic supervision is particularly essential for addressing inconsistencies introduced by generative priors, thereby markedly improving the performance.

| Components | PSNR↑ | SSIM↑ | LPIPS↓ | mPSNR↑ | mSSIM↑ | mLPIPS↓ | FID↓ |
|---|---|---|---|---|---|---|---|
| Full model | **18.58** | **0.585** | **0.325** | **19.67** | **0.720** | **0.240** | **114.16** |
| w/o. panoramic representation | 17.53 | 0.573 | 0.333 | 19.34 | 0.709 | 0.242 | 117.90 |
| w/o. trajectory alignment | 17.91 | 0.575 | 0.332 | 19.49 | 0.717 | **0.240** | 127.18 |
| w/o. view expansion | 17.48 | 0.567 | 0.347 | 19.37 | 0.714 | 0.243 | 120.56 |
| w/o. strategic supervision | 14.13 | 0.309 | 0.538 | 14.72 | 0.487 | 0.438 | 208.78 |

Figure 5: **Visual comparison of ablations.** As exhibited in the red-boxed areas, panoramic representation facilitates the effective extension of views beyond camera boundaries. Trajectory alignment and view expansion contribute to the consistent synthesis of extended views of red-boxed areas. Strategic supervision is pivotal in harmonizing the generative content and the original monocular video, culminating in coherent 4D renderings in the orange-boxed areas.

dataset (Yoon et al., 2020), as shown in Figure 4 and Table 2. It also delivers high-fidelity synthesis on other real-world monocular videos with crowded scenes, heavy occlusion, and complex motion. Please refer to Section B in the supplementary material for details.

## 4.3 ABLATION STUDY

We ablate on various components of our method on Li et al. (2023) in Table 3, 4 and Figure 5, 6.

**Panoramic representation.** Incorporating the panoramic representation significantly facilitates the effective extension of views beyond camera boundaries. By reducing the 6-DoF camera search to a 2D field expansion, the panorama allows the system to generate consistent large-baseline viewpoints beyond observed camera boundaries. This leads to pronounced refinements in Figure 5 and improvements in PSNR, SSIM, and LPIPS in Table 3.

**Trajectory alignment.** The trajectory alignment in Equation 6 plays a crucial role in stabilizing the generative prior. The interpolated trajectory avoids abrupt viewpoint jumps and ensures that newly synthesized viewpoints evolve smoothly in alignment with diffusion priors. Our ablations show that full trajectory adaptation markedly improves PSNR, SSIM, and LPIPS when encompassing the entire image.

**View expansion.** The view-expansion mechanism in Equation 8 merges static content from all timestamps into a unified, viewpoint-consistent background. This ensures more consistent inference for extended views, with obvious improvements in PSNR, SSIM, and LPIPS.

**Strategic supervision.** Our strategic-supervision design in Equation 11 uses invisible-region and unreliable-region masks to selectively apply generative priors only where deserve reconstruction. It proves to be pivotal in reconciling inconsistencies between the generative prior and the original monocular video. Consequently, this supervision strategy enhance the coherence and realism of 4D renderings in Figure 5 and leading to obvious improvement in all metrics in Table 3.

Table 4: **Ablations on trajectory directions $M$, interpolation steps $L$, and SSIM threshold $\epsilon$.**

| $M$ | PSNR↑ | SSIM↑ | LPIPS↓ | $L$ | PSNR↑ | SSIM↑ | LPIPS↓ | $\epsilon$ | PSNR↑ | SSIM↑ | LPIPS↓ |
|---|---|---|---|---|---|---|---|---|---|---|---|
| 2 | 17.74 | 0.576 | 0.345 | 12 | 17.92 | 0.578 | 0.342 | 0.1 | 18.33 | 0.572 | 0.340 |
| 8 | **18.58** | **0.585** | **0.325** | 49 | **18.58** | **0.585** | **0.325** | 0.3 | **18.58** | **0.585** | **0.325** |
| 32 | 18.02 | 0.581 | 0.332 | 196 | 17.65 | 0.576 | 0.347 | 0.5 | 18.10 | 0.563 | 0.348 |

**Hyperparameter sensitivity.** We further analyze sensitivity to trajectory directions $M$, interpolation steps $L$, and SSIM threshold $\epsilon$, as shown in Table 4 and Figure 6. Small $M$ in Equation 5 yields insufficient scene exploration. An excessively large $M$ causes multiple novel trajectories to collide, degrading consistency. For interpolation steps $L$ in Equation 7, too few steps create overly abrupt viewpoint transitions, whereas excessively large $L$ requires multiple rounds of diffusion refinement and introduces inconsistency. Finally, the SSIM threshold $\epsilon$ in Equation 10 regulates the ratio of reliable to unreliable regions. A small $\epsilon$ suppresses the refinement of generative priors, while a large $\epsilon$ causes unnecessary supervision from synthesized frames.

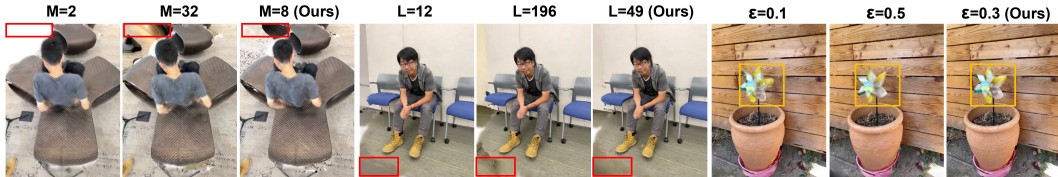

Figure 6: **Ablations on trajectory directions $M$, interpolation steps $L$, and SSIM threshold $\epsilon$.** Trajectory directions $M$ and interpolation steps $L$ has significant influence on red-boxed regions beyond observable boundary. SSIM threshold $\epsilon$ influences detailed textures on orange-boxed regions.

## 5 CONCLUSION

In this paper, we introduce PASTEL, a novel framework that employs Panoramic Alignment for Strategic Exploitation of Generative Priors in monocular 4D reconstruction. Our approach enhances traditional reconstruction-based methods by expanding rendering scopes and overcoming generative models' limitations concerning consistency with monocular video inputs, thus achieving state-of-the-art performance and setting new benchmarks in the field. Central to our approach is the panoramic representation, which strategically selects target cameras and trajectories to optimize the utilization of generative priors, effectively enhancing inference quality in regions beyond observable boundaries. Furthermore, we implement a comprehensive view expansion with strategic 4D scene supervision to mitigate inconsistencies between generative priors and monocular video inputs. We anticipate that the unbounded 4D scene reconstruction achieved by PASTEL will unlock significant potential for immersive Virtual Reality experiences and enriched interactions in embodied AI.

## 6 ETHICS STATEMENT

Our proposed PASTEL enables 4D scene reconstruction with promising applications. As with other video generation methods, there is a risk of misuse for creating misleading or harmful content. In addition, we use TrajectoryCrafter (YU et al., 2025) as a generative prior, which may introduce biases inherited from its training data. We therefore emphasize that any application of our method should carefully consider these risks. They should ensure transparent communication of limitations, and adopt safeguards to prevent malicious or unethical use.

## 7 REPRODUCIBILITY STATEMENT

We are committed to ensuring the reproducibility of our work. To this end, we will release the full implementation and relevant scripts upon the final acceptance of the paper. This timeline allows us to carefully clean and document the code for ease of use. All experimental settings, hyperparameters, and implementation details are described in Section 3 and Section 4.1 to facilitate verification prior to code release.

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

## A    LARGE LANGUAGE MODELS (LLM) USAGE

In preparing this manuscript, we made limited use of Large Language Models (LLMs). Specifically, LLMs were used to correct grammar errors and refine the use of technical terminology to improve readability. We take full responsibility for the content presented in this work.

## B    MORE RESULTS ON REAL-WORLD MONOCULAR VIDEOS

The core motivation of PASTEL is exploration of unseen regions. These regions may involve dynamic objects undergoing complex motion, self-occlusion, or dis-occlusion. Current datasets such as DyCheck (Li et al., 2023) and NSFF (Yoon et al., 2020) rarely contain dynamic content that moves beyond observable boundaries. Moreover, highly dynamic regions fully unobservable cannot be reconstructed reliably by any method. Their motions are completely unobservable from input monocular video. Therefore, we focus our analysis on partially occluded dynamic regions.

To evaluate on the adaptation of PASTEL to other complex real-world monocular videos, we experiment on casual monocular videos of dynamic scenes with irregular motion and crowded environments. As illustrated in Figure 7, PASTEL enables adaptation to irregular movements of the complex hip hop in the first row, heavy occlusion cases of the train in the second row, and crowded environments in the third row. Besides, it is evident that previous reconstruction-based methods (Lei et al., 2025) frequently encounter limitations when attempting to extend beyond observable boundaries. In stark contrast, PASTEL successfully facilitates the generation of novel views while maintaining high fidelity in novel-view synthesis.

We further evaluate PASTEL on highly dynamic scenes with significant occlusions and dis-occlusions. As shown in Figure 8, previous state-of-the-art method (Lei et al., 2025) often produce missing or broken geometry in the occluded moving body parts. In contrast, PASTEL maintains coherent motion trajectories and complements geometry under irregular human motion and heavy occlusions.

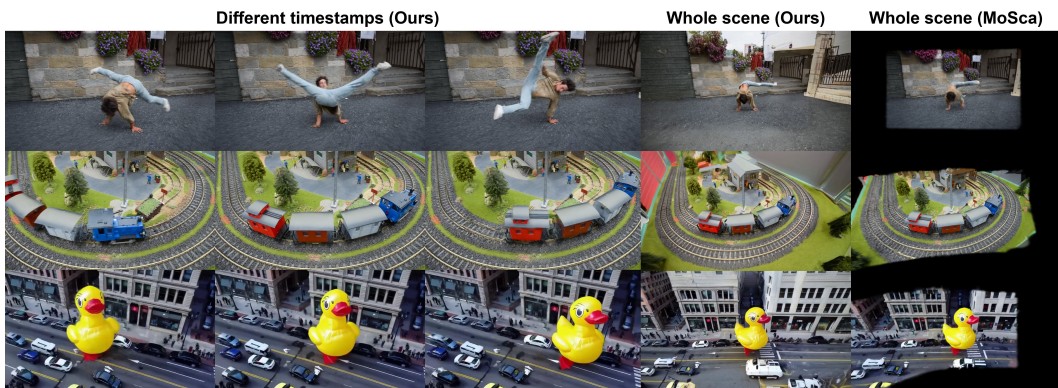

Figure 7: **Reconstruction on real-world monocular videos.** The first three columns exhibit novel view synthesis with different timestamps. The last two columns compare the whole scene with the previous state-of-the-art method, Mosca (Lei et al., 2025). PASTEL can be effectively applied to the reconstruction of any monocular video, achieving both superior overall quality and effective synthesis of extended views.

Table 5: **Ablation study on generative priors.** We compared three recently open-sourced camera-controlled video generation methods. TrajectoryCrafter (YU et al., 2025) performs best in our architecture.

| Generative prior | PSNR↑ | SSIM↑ | LPIPs↓ |
|---|---|---|---|
| TrajectoryCrafter (YU et al., 2025) | **18.58** | **0.585** | **0.325** |
| ReCamMaster (Bai et al., 2025a) | 17.95 | 0.551 | 0.361 |
| ViewCrafter (Yu et al., 2024) | 17.22 | 0.513 | 0.408 |

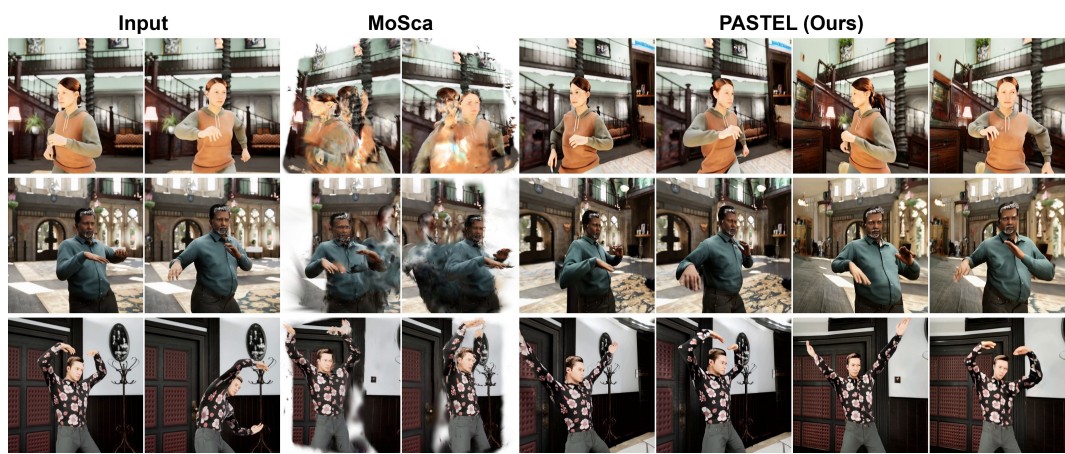

**Input**   **MoSca**   **PASTEL (Ours)**

Figure 8: **Reconstruction on monocular videos with complex motion and heavy occlusions.** PASTEL captures highly dynamic scenes with irregular motion, self-occlusion, and dis-occlusion. Compared with Lei et al. (2025), PASTEL complements geometry of dynamic content in unseen and occluded regions while improving static backgrounds at the same time.

Table 6: **Ablation study on inaccurate depth estimation.** Inaccuracy of depth is simulated with the standard deviation of the original depth.

| Depth std | PSNR↑ | SSIM↑ | LPIPs↓ |
|---|---|---|---|
| 0% | **18.58** | **0.585** | **0.325** |
| 1% | 18.33 | 0.572 | 0.340 |
| 5% | 17.44 | 0.535 | 0.384 |
| 20% | 15.28 | 0.429 | 0.490 |

## C    MORE ABLATIONS

We conduct more ablations of our method on the iPhone dataset (Li et al., 2023).

### C.1    ABLATION ON GENERATIVE PRIORS

Table 5 shows performance variations when using different camera-controlled video generation models as generative priors. Among them, TrajectoryCrafter (YU et al., 2025) proves to be the most effective, likely due to its precise control derived from depth and pose projections. ReCamMaster (Bai et al., 2025a) shows slightly lower performance, possibly because of its deviation in pose estimation. ViewCrafter (Yu et al., 2024) exhibits lower PSNR, since it cannot support dynamic generation. Nevertheless, TrajectoryCrafter (YU et al., 2025) still has certain inconsistencies, as shown in Figure 3. Improving the consistency of generative priors is still crucial for further enhancing our model's performance.

### C.2    SENSITIVITY ANALYSIS ON DEPTH AND POSE PERTURBATIONS

As shown in Table 6 and Table 7, PASTEL maintains strong performance under slight perturbations in depth (1%) and pose (1°) estimation. However, when the estimation errors become significant (20% for depth and 20° for pose), the performance drops noticeably. It illustrates the reliance of PASTEL on accurate depth and pose estimations.

## D    SCENE-SPECIFIC ADAPTATION

To evaluate on scene-specific adaptation, we fine-tune our diffusion prior using only observed regions of the input monocular video. Specifically, we exploit TrajectoryCrafter YU et al. (2025) as generative

Table 7: **Ablation study on inaccurate pose estimation.** Inaccuracy of camera pose is simulated by adding rotation angles of the three axes. The rotation angles are conducted with the standard deviation of the given number.

| Pose std | PSNR↑ | SSIM↑ | LPIPs↓ |
|---|---|---|---|
| 0° | **18.58** | **0.585** | **0.325** |
| 1° | 18.30 | 0.570 | 0.336 |
| 5° | 17.21 | 0.508 | 0.402 |
| 20° | 14.67 | 0.398 | 0.535 |

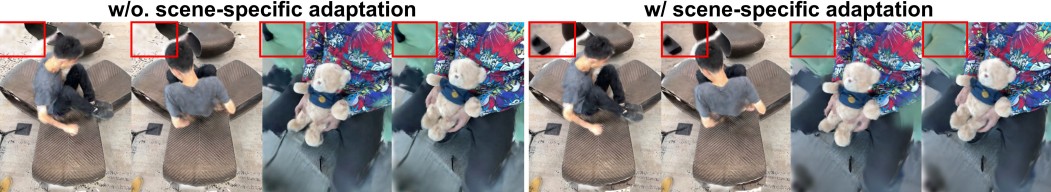

**w/o. scene-specific adaptation**    **w/ scene-specific adaptation**

Figure 9: **Scene-specific adaptation of the generative prior.** Fine-tuning on observed regions show only slight improvement of color consistency in red-boxed unseen regions.

prior. We finetune the cross-attention and patch embedding layers in the Ref-DiT blocks of our generative prior while freezing all other parameters on monocular input videos.

As shown in Figure 9, fine-tuning yielded a slight improvement in color consistency for unseen regions. We hypothesize this is because these unexplored regions remain entirely unobserved in the monocular input. Therefore, scene-specific adaptation cannot provide reliable geometric supervision for them. Moreover, PASTEL already conditions its generative prior on monocular observations. The generative prior have effectively conditioned on the scene without further fine-tuning. Therefore, additional scene-specific finetuning therefore brings limited improvement.

## E FUTURE WORKS

Despite the high fidelity and consistency of our reconstruction results, our method still relies on accurate depth, pose, and flow estimations, as well as stable camera-controlled video generation priors, as shown in Section C.2 and Section C.1. On the other hand, Zhang et al. (2025); Wang et al. (2025a;b) recently proposed feed-forward neural network that directly infers all key 3D attributes of a scene, including depth and pose estimation. However, their requirement for GPU memory is directly proportional to the frame number of the monocular video. Consequently, it is challenging to apply these methods directly to monocular videos. Meanwhile, some co-current works (Wu et al., 2025; Yesiltepe & Yanardag, 2025) recently achieved improved performance on camera-controlled video generation. We also plan to explore these works as generative priors after their code release. Another promising research direction would be to design a feed-forward network for directly reconstructing both observable boundaries and unbounded content beyond observable boundaries. This feed-forward network may be achieved by delicate integration of both feed-forward neural networks (Zhang et al., 2025; Wang et al., 2025a;b) and video generation models (YU et al., 2025; Bai et al., 2025a; Wu et al., 2025; Yesiltepe & Yanardag, 2025).

