# OpenReview forum: "PASTEL: Panoramic Alignment for Monocular 4D Reconstruction"
_ICLR.cc/2026/Conference — Submitted to ICLR 2026_

### Official Review · Reviewer_c8tk · 2025-10-29

**Soundness:** 3
**Presentation:** 3
**Contribution:** 3
**Rating:** 6
**Confidence:** 3

**Summary:**

This paper proposes PASTEL, a panoramic alignment framework for monocular 4D reconstruction that extends scene coverage beyond visible camera regions while maintaining spatial-temporal consistency. The method integrates panoramic alignment through spherical projection, adaptive trajectory identification to regulate camera motion, and strategic 4D supervision to reduce generative inconsistencies when using pre-trained priors such as TrajectoryCrafter. By unifying these modules into a single pipeline, PASTEL improves novel-view fidelity and stability. Experiments on DyCheck and NSFF demonstrate consistent gains in PSNR, SSIM, and perceptual metrics over existing baselines, validating the effectiveness of panoramic alignment for monocular 4D reconstruction.

**Strengths:**

1.This work addresses a clear and under-explored challenge, which is handling “invisible regions” in monocular 4D scenes.

2.Panoramic alignment and trajectory identification are intuitive and practically effective.

3.Comprehensive experiments on DyCheck and NSFF, including covisible and out-of-view regions, with strong PSNR/SSIM/LPIPS/FID performance.

4.Ablation studies cover most modules and show consistent trends.

**Weaknesses:**

1.While the method provides reasonably detailed descriptions of its panoramic trajectory design, view-expansion blending, and strategic-supervision weighting, several aspects of their motivation and behavior remain underexplored. In particular, Equations (6–11) define the optimization objectives but lack intuitive explanation or empirical analysis demonstrating why these components stabilize generative priors. Additional visualization or ablation-based interpretation, for example showing how trajectory adaptation improves consistency, would make the mechanism more convincing.

2.The framework primarily combines existing paradigms, such as spherical reprojection, generative priors (TrajectoryCrafter), and 4DGS/MoSca representations, without introducing fundamentally new architectures or loss designs. While the panoramic alignment component is intuitively motivated, it can be viewed more as a geometric reparameterization than a novel learning mechanism, making the conceptual contribution appear more incremental than transformative.

3.Performance relies heavily on accurate depth/pose/flow predictions and the generative quality of TrajectoryCrafter. While the sensitivity analyses help, this dependency somewhat limits generalizability.

**Questions:**

The paper would benefit from more in-depth clarification of its core mechanisms. In particular, the panoramic trajectory design, view-expansion blending, and strategic-supervision weighting are introduced with equations but lack clear intuition or visualization showing how they stabilize generative priors. It would also help to further distinguish PASTEL from existing generative-alignment frameworks such as TrajectoryCrafter or MoSca, clarifying whether the improvements stem from new algorithmic design or from the integration of known components.

---

> ### Author Response · Authors · 2025-11-26
>
> Thanks for your valuable feedback. As you mentioned, 'This work addresses a clear and under-explored challenge', our proposed panoramic reformulation is 'intuitive and practically effective', with 'comprehensive experiments' and 'strong performance'. We further modified our paper to highlight  in-depth clarification of the core mechanisms and systematic novelty.
>
>
> > **W1:** While the method provides reasonably detailed descriptions of its panoramic trajectory design, view-expansion blending, and strategic-supervision weighting, several aspects of their motivation and behavior remain underexplored. In particular, Equations (6–11) define the optimization objectives but lack intuitive explanation or empirical analysis demonstrating why these components stabilize generative priors. Additional visualization or ablation-based interpretation, for example showing how trajectory adaptation improves consistency, would make the mechanism more convincing.
>
> **R:** Good suggestion. We have expanded the Method section, revised the ablations in Sec. 4.3, and added new visual and quantitative analyses in Table 3,4 and Figure 5,6 for more in-depth clarification.
>
> For clarity, we pasted and arranged the exact motivation and empirical observations added to the paper:
>
> >>**1. Trajectory alignment:**
> >>*Intuition:* This design provides a smooth, globally consistent prior that avoids abrupt view changes. It ensures that novel views align smoothly with diffusion model expectations.
> >>*Empirical analysis:* As shown in Figure 5, the interpolated trajectory in Equation 6 avoids abrupt viewpoint jumps and ensures that newly synthesized viewpoints evolve smoothly in alignment with diffusion priors. Our ablations in Table 3 show that full trajectory adaptation markedly improves PSNR, SSIM, and LPIPS when encompassing the entire image. Table 4 and Figure 6 shows small trajectory direction number $M$ in Equation 5 yields insufficient scene exploration. An excessively large $M$ causes multiple novel trajectories to collide, degrading consistency.
> >>
> >>**2. View expansion:**
> >>*Intuition:* Static content from all timestamps can be merged into a unified, viewpoint-consistent background. It prevents the diffusion model from hallucinating inconsistent geometry and provides a stable global reference across the entire 4D sequence.
> >>*Empirical analysis:* The view-expansion mechanism in Equation 8 merges static content from all timestamps into a unified, viewpoint-consistent background. This ensures more consistent inference for extended views, with obvious improvements in PSNR, SSIM, and LPIPS in Table 3. For interpolation steps $L$ in Equation 7, too few steps create overly abrupt viewpoint transitions, whereas excessively large $L$ requires multiple rounds of diffusion refinement and introduces inconsistency.
> >>
> >>**3. Strategic supervision:**
> >>*Intuition:* Unreliable and invisible masks ensure generative priors are specifically applied to regions that require refinement. It prevents diffusion priors from overwriting well-reconstructed regions. It further avoids inconsistencies arising from generative priors which may artifact the final 4D scene.
> >>*Empirical analysis:* Our strategic-supervision design in Equation 11 uses invisible-region and unreliable-region masks to selectively apply generative priors only where deserve reconstruction. It proves to be pivotal in reconciling inconsistencies between the generative prior and the original monocular video. The SSIM threshold $\epsilon$ in Equation 10 regulates the ratio of reliable to unreliable regions. A small $\epsilon$ suppresses the refinement of generative priors, while a large $\epsilon$ causes unnecessary supervision from synthesized frames.

---

> ### Author Response · Authors · 2025-11-26
>
> > **W2:** The framework primarily combines existing paradigms, such as spherical reprojection, generative priors (TrajectoryCrafter), and 4DGS/MoSca representations, without introducing fundamentally new architectures or loss designs. While the panoramic alignment component is intuitively motivated, it can be viewed more as a geometric reparameterization than a novel learning mechanism, making the conceptual contribution appear more incremental than transformative.
>
> **R:**  Thanks for valuable comments. We expand the discussion of mechanism-level contributions in Lines 76–91 of the Introduction to clarify that our contribution goes beyond incremental geometric reparameterization. We introduce two transformative, mechanism-level reformulations.
>
> 1. **Panoramic representation reformulation.** Existing approaches must optimize (M × L × 4 × 4) parameters for (M) trajectories with (L) viewpoints in full 3D Cartesian pose space. This search space is extremely high-dimensional, unstructured, difficult for exploration, and offers no global structural constraints. In contrast, our approach collapses this large space into a structured 2D directional search space. It transformatively allows scene expansion to be solved as structured trajectories. This is a fundamentally different learning mechanism, not an architectural tweak.
>
> 2. **4D reconstruction reformulation.** Prior 4D reconstruction methods are inherently limited within observable boundaries. Without a global representation, they cannot strategically explore unobserved regions. In comparison, our reformulation is the **first** to enable complete reconstruction beyond observable boundaries. This represents a transformative shift in how monocular 4D synthesis is conceptualized.
>
>
> > **W3:** Performance relies heavily on accurate depth/pose/flow predictions and the generative quality of TrajectoryCrafter. While the sensitivity analyses help, this dependency somewhat limits generalizability.
>
> **R:**  Thanks for this important observation. Our experiments did not reveal instability from these dependencies, but we agree this is a key factor for robustness.  We highlight two mechanisms that could mitigate such issues:
>
> 1. Robustness to Imperfect Geometry: Our pipeline is compatible with alternative geometry sources. Notably, depth/pose can be obtained directly from the dynamic Gaussian representation. This could ensure intrinsic consistency while reduce reliance on external estimators.
>
> 2. Robustness to Generative Prior Quality: We have designed parameter $\epsilon$ to regulate the influence of the generative prior. We can lower $\epsilon$ to effectively reduce the impact of generative prior. Setting $\epsilon=0$ allows our method to fall back to a purely reconstruction-based mode, entirely avoiding any potential negative impact from the generative prior. This allows our system to adapt to cases where external priors may be unreliable.
>
> > **Q:** The paper would benefit from more in-depth clarification of its core mechanisms. In particular, the panoramic trajectory design, view-expansion blending, and strategic-supervision weighting are introduced with equations but lack clear intuition or visualization showing how they stabilize generative priors. It would also help to further distinguish PASTEL from existing generative-alignment frameworks such as TrajectoryCrafter or MoSca, clarifying whether the improvements stem from new algorithmic design or from the integration of known components.
>
> **R:**  Please refer to response to **W1** and  **W2** for in-depth clarification and paper modification about our core mechanisms.
>
> **Existing frameworks**  (TrajectoryCrafter, MoSca) operate strictly in Cartesian pose space. They are fundamentally unable to perform whole-scene exploration beyond observable regions. As shown in Fig. 3, TrajectoryCrafter produces severe geometric distortion. MoSca conducts reconstruction limited on observed regions.
>
> Even integration of both components cannot achieve whole 4D scene exploration. Direct integration will result in low efficiency and severe inconsistency. For example, in Figure 3, even one inference of TrajectoryCrafter achieves severe distortion. Naive integration with multiple inferences of TrajectoryCrafter through random trajectories will only lead to worse distortion and inconsistency, completely disrupting the scene.
>
> In comparison, our method reformulates 4D synthesis in a panoramic domain. It significantly reduces the degrees of freedom and enables stable exploration to unseen regions. Therefore, our core contribution is not a direct integration of existing modules, but a new panoramic reformulation. You may refer to the reply to **W2** in detail.

---

### Official Review · Reviewer_wtCL · 2025-11-01

**Soundness:** 2
**Presentation:** 2
**Contribution:** 2
**Rating:** 4
**Confidence:** 4

**Summary:**

The paper “PASTEL: Panoramic Alignment for Monocular 4D Reconstruction” proposes a framework that reconstructs 4D dynamic scenes from monocular videos by integrating panoramic alignment and generative priors. The method introduces a spherical panoramic representation to design optimal camera trajectories that balance deviation minimization and exploration beyond visible regions. It then employs a static-dynamic projection strategy to expand views and uses diffusion-based video generation (TrajectoryCrafter) as a prior for inpainting unseen areas. The approach achieves competitive results on DyCheck and NSFF, outperforming reconstruction- and generation-based baselines.

**Strengths:**

+ The paper tackles an important challenge: recovering unobserved 4D regions from monocular videos, by leveraging generative diffusion priors in a geometrically guided way.
+ The paper provides extensive evaluations on both synthetic and real-world datasets, ablation studies on panoramic alignment, trajectory alignment, and supervision masks, and sensitivity analyses on depth/pose errors.

**Weaknesses:**

- Overly complicated and fragmented pipeline. The method involves multiple non-trivial steps: panoramic reprojection, trajectory identification, static-dynamic decomposition, diffusion-based video generation, Gaussian splatting refinement, and strategic supervision, which together make the system difficult to apply and obscure the core technical novelty.

- Limited novelty beyond careful integration. While the panoramic alignment and strategic supervision are well-engineered, the approach mainly repackages existing components rather than introducing fundamentally new learning mechanisms or representations. The contribution feels more architectural than conceptual.

- Weak justification and dependency on external priors. The framework heavily depends on pretrained generative priors and external depth/pose estimators (UniDepth, RAFT, Bootstapir). This reliance diminishes the originality and raises concerns about how much of the improvement comes from the proposed framework versus stronger pretrained modules.

**Questions:**

Could the proposed panoramic alignment be simplified or integrated into an end-to-end learning model rather than relying on handcrafted reprojection and trajectory rules?

---

> ### Author Response · Authors · 2025-11-26
>
> Thank you for constructive comments. As you have mentioned, this paper 'tackles an important challenge' with 'extensive evaluations'. In the revision, we have reduced overly architectural descriptions, clarified the conceptual contribution, and emphasized our core novelty.
>
> > **W1:** Overly complicated and fragmented pipeline. The method involves multiple non-trivial steps: panoramic reprojection, trajectory identification, static-dynamic decomposition, diffusion-based video generation, Gaussian splatting refinement, and strategic supervision, which together make the system difficult to apply and obscure the core technical novelty.
>
> **R:**  Thank you for the helpful suggestion. We have revised  both the Introduction and Method sections to explicitly highlight our core technical novelty.
>
> As updated in Lines 76-85, our core technical contribution is the panoramic reformulation, which transforms the 4D scene reconstruction into a panoramic space. This removes the need to design high-dimensional camera extrinsics in full 3D Cartesian space, which often leads to suboptimal coverage and cross-view inconsistencies. Our reformulation makes trajectory expansion tractable, interpretable, and globally consistent.
>
> As modified in Lines 204-209, without our reformulation, one must optimize (M × L × 4 × 4) parameters for (M) trajectories with (L) viewpoints each. It is extremely difficult to ensure global scene coverage and avoid inconsistent or redundant viewpoints. This is the core reason why previous works cannot achieve 4D scene synthesis beyond observable boundaries.
>
> In contrast, our panoramic parameterization collapses the high-dimensional trajectory space into a compact spherical coordinate system.  As revised in Lines 230–235, this structured representation greatly simplifies optimization and explicitly exposes global scene geometry. It directly leads to efficient and consistent novel-view expansion.
>
>
> > **W2:** Limited novelty beyond careful integration. While the panoramic alignment and strategic supervision are well-engineered, the approach mainly repackages existing components rather than introducing fundamentally new learning mechanisms or representations. The contribution feels more architectural than conceptual.
>
>
> **R:** We apologize for confusion. We have revised the introduction to clearly articulate that our core contribution is conceptual, not architectural. The architectural components you mentioned exist only to execute our conceptual reformulation.
>
>
> As modified in Lines 86-91, our core conceptual contributions are:
>
> 1. **Panoramic representation reformulation.** Existing approaches must optimize (M × L × 4 × 4) parameters for (M) trajectories with (L) viewpoints in full 3D Cartesian pose space. This search space is extremely high-dimensional, unstructured, difficult for exploration, and offers no global structural constraints. In contrast, our approach collapses this large space into a structured 2D directional search space. It transformatively allows scene expansion to be solved as structured trajectories. This is a fundamentally different learning mechanism, not an architectural tweak.
>
> 2. **4D reconstruction reformulation.** Prior 4D reconstruction methods are inherently limited within observable boundaries. Without a global representation, they cannot strategically explore unobserved regions. In comparison, our reformulation is the **first** to enable complete reconstruction beyond observable boundaries. This represents a transformative shift in how monocular 4D synthesis is conceptualized.

---

> ### Author Response · Authors · 2025-11-26
>
> > **W3:** Weak justification and dependency on external priors. The framework heavily depends on pretrained generative priors and external depth/pose estimators (UniDepth, RAFT, Bootstapir). This reliance diminishes the originality and raises concerns about how much of the improvement comes from the proposed framework versus stronger pretrained modules.
>
> **R:** This is a misunderstanding. All baselines, including Mosca and TrajectoryCrafter, rely on similar pretrained depth/pose estimators (UniDepth, RAFT, Bootstapir) and generative priors. Our comparisons share similar pretrained modules. Therefore, improvements cannot come from stronger external priors.
>
> Our core contribution, the panoramic alignment representation, is completely independent of generative priors and external depth/pose estimators. The observed performance gains arise directly from the proposed reformulation, not from pretrained components shared across all baselines.
>
>
> > **Q:** Could the proposed panoramic alignment be simplified or integrated into an end-to-end learning model rather than relying on handcrafted reprojection and trajectory rules?
>
> **R:** Yes. Our core contribution, panoramic alignment representation, is model-agnostic and are compatible with any end-to-end learning model.
>
> Specifically, a possible integration is as follows:
>
> 1. Use our panoramic formulation to predefine globally consistent trajectories to ensure full scene coverage.
> 2. Warp input video frames or learned feature maps into the panoramic trajectory space
> 3. Feed the warped sequences into any end-to-end backbone (e.g., VGGT for pose/depth estimation, Lyra for Gaussian primitives).
>
> Note that our panoramic representation and warping operations are computationally inexpensive and introduce minimal overhead. Thus, they are fully compatible with end-to-end learning.

---

> > ### Comment · Reviewer_wtCL · 2025-11-26
> >
> > I appreciate the author's detailed rebuttal, which well addressed most of my concerns. I have also read the revised manuscript, and it shows much better structure and presentation for me. Thus, I would like to raise my rating to 6.

---

### Official Review · Reviewer_pwmL · 2025-11-01

**Soundness:** 3
**Presentation:** 3
**Contribution:** 3
**Rating:** 6
**Confidence:** 3

**Summary:**

This paper presents PASTEL, a method for reconstructing geometry- and motion-consistent 4D scenes from monocular video that explicitly addresses the challenging problem of synthesizing content in regions outside the observed camera coverage. The key innovation lies in transforming the reconstruction task into panorama space, which simplifies novel trajectory design and enables strategic use of generative priors to fill invisible regions while maintaining geometric consistency.

**Strengths:**

- PASTEL introduces a novel framework for monocular 4D scene reconstruction that not only rebuilds visible regions but also consistently infers "invisible regions" beyond observable camera limits.
- Also, the elegant dimensional reduction is quite interesting.  The panorama alignment is a genuinely clever contribution that reduces the high-dimensional camera extrinsic parameter search to a tractable 2D directional search using dm=(cos⁡(2πm/M),sin⁡(2πm/M))d_m = (\cos(2\pi m/M), \sin(2\pi m/M)), dm​=(cos(2πm/M),sin(2πm/M)). This makes trajectory sampling computationally feasible and interpretable.
- The method leverages a panoramic alignment to identify optimal camera trajectories for generative priors, followed by a static-dynamic projection and strategic 4D scene supervision to mitigate inconsistencies.

**Weaknesses:**

1. Generative prior domain shift: Diffusion models trained on large-scale datasets can introduce stylistic and illumination inconsistencies that conflict with the specific scene.
- Have you experimented with scene-specific adaptation of the generative prior (e.g., LoRA fine-tuning on observed regions)? If so, what was the impact on generalization to unseen regions?

2. Hyperparameter sensitivity: The method introduces several hyperparameters (MM M for trajectory directions, LL L for interpolation steps, ϵ\epsilon ϵ for SSIM threshold).
-  How sensitive are results to these choices? Is there an automated or principled way to select them?

3. Performance gains primarily from static regions: While the direct warping approach is novel, a critical concern is that the reported performance improvements appear to be predominantly driven by static background regions rather than dynamic content.
 If the improvement is mostly in static regions, the contribution becomes less compelling for true 4D scene understanding, as static region reconstruction from multiple views is a more well-studied problem. The value proposition should be in handling dynamic content in unseen regions.

4. Limited analysis of dynamic content generation: The core challenge in 4D reconstruction is handling dynamic objects with complex motion.
- How well does the method generalize to highly dynamic scenes with significant occlusions and dis-occlusions? The provided examples seem to feature relatively simple motions.

**Questions:**

N/A (Included in weakness)

---

> ### Author Response · Authors · 2025-11-26
>
> Thanks for your insightful feedback and for recognizing the novelty of our contribution. As you mentioned, our panoramic alignment representation “transforms the reconstruction task into panorama space, reducing high-dimensional search into 2D trajectory identification.”
>
>
> > **W1:** Generative prior domain shift: Diffusion models trained on large-scale datasets can introduce stylistic and illumination inconsistencies that conflict with the specific scene.
> Have you experimented with scene-specific adaptation of the generative prior (e.g., LoRA fine-tuning on observed regions)? If so, what was the impact on generalization to unseen regions?
>
> **R:** Thanks for insightful advice. We have added additional experiments on scene-specific adaptation in Section D in supplementary. These experiments show a slight improvement in color consistency in unseen regions. As clarified in the paper, we now state:
>
> > > **Experiments:** To evaluate scene-specific adaptation, we fine-tune our diffusion prior using only observed regions of the input monocular video. We exploit TrajectoryCrafter as generative prior. We finetune the cross-attention and patch embedding layers in the Ref-DiT blocks of our generative prior while freezing all other parameters on monocular input videos.
> > > **Results:** Scene-specific fine-tuning yielded a slight improvement in color consistency for unseen regions. We hypothesize this is because these unexplored regions remain entirely unobserved in the monocular input. Therefore, scene-specific adaptation cannot provide reliable geometric supervision for them. Moreover, our model already conditions its generative prior on monocular observations. The generative prior has effectively conditioned on the scene without further fine-tuning. Therefore, additional scene-specific finetuning therefore brings limited improvement.
>
>
>
> > **W2:** Hyperparameter sensitivity: The method introduces several hyperparameters (MM M for trajectory directions, LL L for interpolation steps, ϵ\epsilon ϵ for SSIM threshold).
> How sensitive are results to these choices? Is there an automated or principled way to select them?
>
> **R:** Good point. We provide additional hyperparameter sensitivity analysis in Table 4, Figure 6, and Section 4.3. All hyperparameters are selected through grid search experiments. Our method is not overly sensitive to these hyperparameter choices. As stated in the paper:
>
> >>**Results:** Small $M$ yields insufficient scene exploration. An excessively large $M$ causes multiple novel trajectories to collide, reducing consistency.
> For interpolation steps $L$, too few steps create overly abrupt viewpoint transitions, whereas excessively large $L$ requires multiple rounds of diffusion refinement and introduces inconsistency.
> Finally, the SSIM threshold $\epsilon$ regulates the ratio of reliable to unreliable regions. A small $\epsilon$ suppresses the refinement of the generative priors, while a large $\epsilon$ causes unnecessary supervision of the synthesized frames.
>
> | $M$   | PSNR$\uparrow$ | SSIM$\uparrow$ | LPIPS$\downarrow$ |
> |:------|:---------------|:---------------|:------------------|
> | 2     | 17.74          | 0.576          | 0.345             |
> | 8     | **18.58**      | **0.585**      | **0.325**         |
> | 32    | 18.02          | 0.581          | 0.332             |
>
> | $L$   | PSNR$\uparrow$ | SSIM$\uparrow$ | LPIPS$\downarrow$ |
> |:------|:---------------|:---------------|:------------------|
> | 12    | 17.92          | 0.578          | 0.342             |
> | 49    | **18.58**      | **0.585**      | **0.325**         |
> | 196   | 17.65          | 0.576          | 0.347             |
>
> | $\epsilon$ | PSNR$\uparrow$ | SSIM$\uparrow$ | LPIPS$\downarrow$ |
> |:-----------|:---------------|:---------------|:------------------|
> | 0.1        | 18.33          | 0.572          | 0.340             |
> | 0.3        | **18.58**      | **0.585**      | **0.325**         |
> | 0.5        | 18.10          | 0.563          | 0.348             |

---

> ### Author Response · Authors · 2025-11-26
>
> > **W3:** Performance gains primarily from static regions: While the direct warping approach is novel, a critical concern is that the reported performance improvements appear to be predominantly driven by static background regions rather than dynamic content. If the improvement is mostly in static regions, the contribution becomes less compelling for true 4D scene understanding, as static region reconstruction from multiple views is a more well-studied problem. The value proposition should be in handling dynamic content in unseen regions.
>
>
>
> In these settings, prior state-of-the-art methods (e.g., MoSca) frequently produce missing or fragmented geometry in occluded moving body parts. In contrast, our method maintains coherent motion trajectories and successfully completes geometry even under heavy occlusions. These new evaluations demonstrate that our improvements are not limited to static backgrounds, but extend to challenging dynamic content.
>
> **R:** Excellent suggestion. We have added more challenging, highly dynamic examples in Figure 8 and Section B of the supplementary.
>
> We emphasize that fully unobservable dynamic regions cannot be reliably reconstructed by any method. Their motions are never captured and are completely unobservable. Therefore, we evaluated partially occluded dynamic regions, where meaningful reconstruction is possible.
>
> In these settings, prior state-of-the-art methods (e.g., MoSca) frequently produce missing or fragmented geometry in occluded moving body parts. In contrast, our method maintains coherent motion trajectories and successfully completes geometry even under heavy occlusions. These new evaluations demonstrate that our improvements are not limited to static backgrounds, but extend to challenging dynamic content.
>
> > **W4:** Limited analysis of dynamic content generation: The core challenge in 4D reconstruction is handling dynamic objects with complex motion.
> How well does the method generalize to highly dynamic scenes with significant occlusions and dis-occlusions? The provided examples seem to feature relatively simple motions.
>
> **R:** Good point. Our original submission included results with irregular motions and crowded interaction scenes (Supplementary Figure 7). To further strengthen this evaluation, we now introduce additional highly dynamic scenes with significant occlusions and dis-occlusions in Figure 8.
>
> These new results show that competing approaches (e.g., MoSca) often fail to reconstruct body parts that undergo occlusion and dis-occlusion. By contrast, our method maintains stable, temporally consistent, and complete reconstructions in these challenging scenarios. These experiments demonstrate that our model generalizes robustly to highly dynamic scenes, not only to simple motions.

---

### Author Response · Authors · 2025-12-03
**General Rebuttal Comment**

We sincerely appreciate all constructive comments. To assist final evaluation, we summarize our recognized strengths and paper revisions in response.

## Recognized Strengths:

**4D reconstruction reformulation.** All reviewers agree that this paper 'addresses a clear and under-explored challenge,  handling invisible regions in monocular 4D scenes.'

**Panoramic representation reformulation.** Our proposed panoramic reformulation was consistently regarded as elegant, novel and impactful. Reviewer pwmL highlighted our 'genuinely clever contribution that reduces the high-dimensional camera extrinsic parameter search to a tractable 2D directional search'. Reviewer c8tk described our approach as 'intuitive and practically effective'.

**Strong Performance**. Reviewers agreed that the paper presents 'comprehensive experiments with strong performance'.

## Revisions:

**Reviewer pwmL** (Initial Score: 6)

We fully addressed all concerns through
1) Scene-specific adaptation experiments (Supplementary Sec. D).
2)  Hyperparameter sensitivity analysis (Table 4, Fig. 6, Sec. 4.3).
3)  More challenging dynamic scenes with substantial occlusion/dis-occlusion (Fig. 8, Supplementary Sec. B).


**Reviewer wtCL** (Initial Score: 4, raised to 6 before leakage)

We have revised the Introduction and Method sections. Our revision reduced overly architectural descriptions, clarified the conceptual contribution, and emphasized our core novelty. Reviewer wtCL finally remarked:
>I appreciate the author's detailed rebuttal, which well addressed most of my concerns. I have also read the revised manuscript, and it shows much better structure and presentation for me. Thus, I would like to raise my rating to 6.

**Reviewer c8tk** (Initial Score: 6)

We expanded the Method section, revised the ablations in Sec. 4.3, and added new visual and quantitative analyses in Table 3,4 and Figure 5,6 for more in-depth clarification of our methods.

---

### Meta-Review · Area_Chair_jwKj · 2026-01-07

**Summary:**

PASTEL proposes monocular 4D reconstruction that expands beyond the input camera coverage by reprojecting the video into a spherical panorama and reducing trajectory search to a compact 2D directional space, then using a trajectory-guided video diffusion prior plus selective supervision to inpaint unseen regions and refine the 4D representation. Strengths are the panorama formulation and solid evaluations; weaknesses are a complex multi-stage system, heavy reliance on pretrained priors and external geometry, and borderline novelty. Reviewers are generally lukewarm.

**Reviewer Concerns:**

Addressed: added scene-specific adaptation results, hyperparameter sensitivity, more challenging dynamic/occlusion examples, and clearer mechanism explanations; wtCL explicitly said most concerns were resolved and raised the score.

Still outstanding: pipeline complexity, dependence on TrajectoryCrafter plus depth/pose/flow, and novelty feeling like careful integration rather than a clear new learning method; evidence on truly hard dynamic unseen motion remains limited.

**Reviewer Scores:**

pwmL: likely unchanged at 6, still borderline and explicitly fine with rejection.

wtCL: moved 4 -> 6 after rebuttal and revision; would stay at 6 in full discussion.

c8tk: likely unchanged at 6, still views novelty as incremental and notes dependence on external predictions and the generative prior.

Net effect: small upward shift, but weak enthusiasm.

---

### Decision · Program_Chairs · 2026-01-26

Reject